# STOCHASTIC GRADIENT DESCENT PERFORMS VARIATIONAL INFERENCE, CONVERGES TO LIMIT CYCLES FOR DEEP NETWORKS

**Pratik Chaudhari, Stefano Soatto**

Computer Science, University of California, Los Angeles.

Email: pratikac@ucla.edu, soatto@ucla.edu

## ABSTRACT

Stochastic gradient descent (SGD) is widely believed to perform implicit regularization when used to train deep neural networks, but the precise manner in which this occurs has thus far been elusive. We prove that SGD minimizes an average potential over the posterior distribution of weights along with an entropic regularization term. This potential is however not the original loss function in general. So SGD does perform variational inference, but for a different loss than the one used to compute the gradients. Even more surprisingly, SGD does not even converge in the classical sense: we show that the most likely trajectories of SGD for deep networks do not behave like Brownian motion around critical points. Instead, they resemble closed loops with deterministic components. We prove that such "out-of-equilibrium" behavior is a consequence of highly non-isotropic gradient noise in SGD; the covariance matrix of mini-batch gradients for deep networks has a rank as small as 1% of its dimension. We provide extensive empirical validation of these claims, proven in the appendix.

**Keywords:** deep networks, stochastic gradient descent, variational inference, gradient noise, out-of-equilibrium, thermodynamics, Wasserstein metric, Fokker-Planck equation, wide minima, Markov chain Monte Carlo

## 1 INTRODUCTION

Our first result is to show precisely in what sense stochastic gradient descent (SGD) implicitly performs variational inference, as is often claimed informally in the literature. For a loss function $f(x)$ with weights $x \in \mathbb{R}^d$, if $\rho^{\text{ss}}$ is the steady-state distribution over the weights estimated by SGD,

$$\rho^{\text{ss}} = \arg\min_{\rho} \ \mathbb{E}_{x \sim \rho}\left[\Phi(x)\right] - \frac{\eta}{2b}\, H(\rho);$$

where $H(\rho)$ is the entropy of the distribution $\rho$ and $\eta$ and $b$ are the learning rate and batch-size, respectively. The potential $\Phi(x)$, which we characterize explicitly, is related but not necessarily equal to $f(x)$. It is only a function of the architecture and the dataset. This implies that SGD implicitly performs variational inference with a uniform prior, albeit of a different loss than the one used to compute back-propagation gradients.

We next prove that the implicit potential $\Phi(x)$ is equal to our chosen loss $f(x)$ if and only if the noise in mini-batch gradients is isotropic. This condition, however, is *not* satisfied for deep networks. Empirically, we find gradient noise to be highly non-isotropic with the rank of its covariance matrix being about 1% of its dimension. Thus, SGD on deep networks implicitly discovers locations where $\nabla\Phi(x) = 0$, these are not the locations where $\nabla f(x) = 0$. This is our second main result: the most likely locations of SGD are not the local minima, nor the saddle points, of the original loss. The deviation of these critical points, which we compute explicitly scales linearly with $\eta/b$ and is typically large in practice.

When mini-batch noise is non-isotropic, SGD does not even converge in the classical sense. We prove that, instead of undergoing Brownian motion in the vicinity of a critical point, trajectories have a deterministic component that causes SGD to traverse closed loops in the weight space. We detect such loops using a Fourier analysis of SGD trajectories. We also show through an example that SGD with non-isotropic noise can even converge to stable limit cycles around saddle points.

## 2 BACKGROUND ON CONTINUOUS-TIME SGD

Stochastic gradient descent performs the following updates while training a network $x_{k+1} = x_k - \eta \, \nabla f_b(x_k)$ where $\eta$ is the learning rate and $\nabla f_b(x_k)$ is the average gradient over a mini-batch $b$,

$$\nabla f_b(x) = \frac{1}{b} \sum_{k \in b} \nabla f_k(x). \tag{1}$$

We overload notation $b$ for both the set of examples in a mini-batch and its size. We assume that weights belong to a compact subset $\Omega \subset \mathbb{R}^d$, to ensure appropriate boundary conditions for the evolution of steady-state densities in SGD, although all our results hold without this assumption if the loss grows unbounded as $\|x\| \to \infty$, for instance, with weight decay as a regularizer.

**Definition 1 (Diffusion matrix $D(x)$).** If a mini-batch is sampled with replacement, we show in Appendix A.1 that the variance of mini-batch gradients is $\text{var}\,(\nabla f_b(x)) = \frac{D(x)}{b}$ where

$$D(x) = \left( \frac{1}{N} \sum_{k=1}^{N} \nabla f_k(x) \, \nabla f_k(x)^\top \right) - \nabla f(x) \, \nabla f(x)^\top \succeq 0. \tag{2}$$

Note that $D(x)$ is independent of the learning rate $\eta$ and the batch-size $b$. It only depends on the weights $x$, architecture and loss defined by $f(x)$, and the dataset. We will often discuss two cases: isotropic diffusion when $D(x)$ is a scalar multiple of identity, independent of $x$, and non-isotropic diffusion, when $D(x)$ is a general function of the weights $x$.

We now construct a stochastic differential equation (SDE) for the discrete-time SGD updates.

**Lemma 2 (Continuous-time SGD).** *The continuous-time limit of SGD is given by*

$$dx(t) = -\nabla f(x) \, dt + \sqrt{2\beta^{-1} D(x)} \, dW(t); \tag{3}$$

*where $W(t)$ is Brownian motion and $\beta$ is the inverse temperature defined as $\beta^{-1} = \frac{\eta}{2b}$. The steady-state distribution of the weights $\rho(z,t) \propto \mathbb{P}(x(t) = z)$, evolves according to the Fokker-Planck equation (Risken, 1996, Ito form):*

$$\frac{\partial \rho}{\partial t} = \nabla \cdot \left( \nabla f(x) \, \rho + \beta^{-1} \, \nabla \cdot \left( D(x) \, \rho \right) \right) \tag{FP}$$

*where the notation $\nabla \cdot v$ denotes the divergence $\nabla \cdot v = \sum_i \partial_{x_i} v_i(x)$ for any vector $v(x) \in \mathbb{R}^d$; the divergence operator is applied column-wise to matrices such as $D(x)$.*

We refer to Li et al. (2017b, Thm. 1) for the proof of the convergence of discrete SGD to (3). Note that $\beta^{-1}$ completely captures the magnitude of noise in SGD that depends only upon the learning rate $\eta$ and the mini-batch size $b$.

**Assumption 3 (Steady-state distribution exists and is unique).** We assume that the steady-state distribution of the Fokker-Planck equation (FP) exists and is unique, this is denoted by $\rho^{ss}(x)$ and satisfies,

$$0 = \frac{\partial \rho^{ss}}{\partial t} = \nabla \cdot \left( \nabla f(x) \, \rho^{ss} + \beta^{-1} \, \nabla \cdot \left( D(x) \, \rho^{ss} \right) \right). \tag{4}$$

## 3    SGD PERFORMS VARIATIONAL INFERENCE

Let us first implicitly define a potential $\Phi(x)$ using the steady-state distribution $\rho^{\text{ss}}$:

$$\Phi(x) = -\beta^{-1} \log \rho^{\text{ss}}(x), \tag{5}$$

up to a constant. The potential $\Phi(x)$ depends only on the full-gradient and the diffusion matrix; see Appendix C for a proof. It will be made explicit in Section 5. We express $\rho^{\text{ss}}$ in terms of the potential using a normalizing constant $Z(\beta)$ as

$$\rho^{\text{ss}}(x) = \frac{1}{Z(\beta)} \, e^{-\beta \Phi(x)} \tag{6}$$

which is also the steady-state solution of

$$dx = \beta^{-1} \, \nabla \cdot D(x) \, dt - D(x) \, \nabla \Phi(x) \, dt + \sqrt{2\beta^{-1} D(x)} \, dW(t) \tag{7}$$

as can be verified by direct substitution in (FP).

The above observation is very useful because it suggests that, if $\nabla f(x)$ can be written in terms of the diffusion matrix and a gradient term $\nabla \Phi(x)$, the steady-state distribution of this SDE is easily obtained. We exploit this observation to rewrite $\nabla f(x)$ in terms a term $D \, \nabla \Phi$ that gives rise to the above steady-state, the spatial derivative of the diffusion matrix, and the remainder:

$$j(x) = -\nabla f(x) + D(x) \, \nabla \Phi(x) - \beta^{-1} \nabla \cdot D(x), \tag{8}$$

interpreted as the part of $\nabla f(x)$ that cannot be written as $D \, \Phi'(x)$ for some $\Phi'$. We now make an important assumption on $j(x)$ which has its origins in thermodynamics.

**Assumption 4 (Force $j(x)$ is conservative).**   We assume that

$$\nabla \cdot j(x) = 0. \tag{9}$$

The Fokker-Planck equation (FP) typically models a physical system which exchanges energy with an external environment (Ottinger, 2005; Qian, 2014). In our case, this physical system is the gradient dynamics $\nabla \cdot (\nabla f \, \rho)$ while the interaction with the environment is through the term involving temperature: $\beta^{-1} \nabla \cdot (\nabla \cdot (D\rho))$. The second law of thermodynamics states that the entropy of a system can never decrease; in Appendix B we show how the above assumption is sufficient to satisfy the second law. We also discuss some properties of $j(x)$ in Appendix C that are a consequence of this. The most important is that $j(x)$ is always orthogonal to $\nabla \rho^{\text{ss}}$. We illustrate the effects of this assumption in Example 19.

This leads us to the main result of this section.

**Theorem 5 (SGD performs variational inference).**   *The functional*

$$F(\rho) = \beta^{-1} \, \text{KL}\big(\rho \, || \, \rho^{\text{ss}}\big) \tag{10}$$

*decreases monotonically along the trajectories of the Fokker-Planck equation (FP) and converges to its minimum, which is zero, at steady-state. Moreover, we also have an energetic-entropic split*

$$F(\rho) = \mathbb{E}_{x \in \rho} \left[ \Phi(x) \right] - \beta^{-1} H(\rho) + \text{constant}. \tag{11}$$

Theorem 5, proven in Appendix F.1, shows that SGD implicitly minimizes a combination of two terms: an "energetic" term, and an "entropic" term. The first is the average potential over a distribution $\rho$. The steady-state of SGD in (6) is such that it places most of its probability mass in regions of the parameter space with small values of $\Phi$. The second shows that SGD has an implicit bias towards solutions that maximize the entropy of the distribution $\rho$.

Note that the energetic term in (11) has potential $\Phi(x)$, instead of $f(x)$. This is an important fact and the crux of this paper.

**Lemma 6 (Potential equals original loss iff isotropic diffusion).** *If the diffusion matrix $D(x)$ is isotropic, i.e., a constant multiple of the identity, the implicit potential is the original loss itself*

$$D(x) = c\, I_{d \times d} \quad \Leftrightarrow \quad \Phi(x) = f(x). \tag{12}$$

This is proven in Appendix F.2. The definition in (8) shows that $j \neq 0$ when $D(x)$ is non-isotropic. This results in a deterministic component in the SGD dynamics which does not affect the functional $F(\rho)$, hence $j(x)$ is called a "conservative force." The following lemma is proven in Appendix F.3.

**Lemma 7 (Most likely trajectories of SGD are limit cycles).** *The force $j(x)$ does not decrease $F(\rho)$ in (11) and introduces a deterministic component in SGD given by*

$$\dot{x} = j(x). \tag{13}$$

*The condition $\nabla \cdot j(x) = 0$ in Assumption 4 implies that most likely trajectories of SGD traverse closed trajectories in weight space.*

### 3.1 Wasserstein gradient flow

Theorem 5 applies for a general $D(x)$ and it is equivalent to the celebrated JKO functional (Jordan et al., 1997) in optimal transportation (Santambrogio, 2015; Villani, 2008) if the diffusion matrix is isotropic. Appendix D provides a brief overview using the heat equation as an example.

**Corollary 8 (Wasserstein gradient flow for isotropic noise).** *If $D(x) = I$, trajectories of the Fokker-Planck equation (FP) are gradient flow in the Wasserstein metric of the functional*

$$F(\rho) = \mathbb{E}_{x \sim \rho}\Big[f(x)\Big] - \beta^{-1} H(\rho). \tag{JKO}$$

Observe that the energetic term contains $f(x)$ in Corollary 8. The proof follows from Theorem 5 and Lemma 6, see Santambrogio (2017) for a rigorous treatment of Wasserstein metrics. The JKO functional above has had an enormous impact in optimal transport because results like Theorem 5 and Corollary 8 provide a way to modify the functional $F(\rho)$ in an interpretable fashion. Modifying the Fokker-Planck equation or the SGD updates directly to enforce regularization properties on the solutions $\rho^{\text{ss}}$ is much harder.

### 3.2 Connection to Bayesian inference

Note the absence of any prior in (11). On the other hand, the evidence lower bound (Kingma and Welling, 2013) for the dataset $\Xi$ is,

$$
\begin{aligned}
-\log p(\Xi) &\leq \mathbb{E}_{x \sim q}\big[f(x)\big] + \mathrm{KL}\Big(q(x\,|\,\Xi) \,||\, p(x\,|\,\Xi)\Big), \\
&\leq \mathbb{E}_{x \sim q}\big[f(x)\big] - H(q) + H(q, p);
\end{aligned}
\tag{ELBO}
$$

where $H(q, p)$ is the cross-entropy of the estimated steady-state and the variational prior. The implicit loss function of SGD in (11) therefore corresponds to a uniform prior $p(x\,|\,\Xi)$. In other words, we have shown that SGD itself performs variational optimization with a uniform prior. Note that this prior is well-defined by our hypothesis of $x \in \Omega$ for some compact $\Omega$.

It is important to note that SGD implicitly minimizes a potential $\Phi(x)$ instead of the original loss $f(x)$ in ELBO. We prove in Section 5 that this potential is quite different from $f(x)$ if the diffusion matrix $D$ is non-isotropic, in particular, with respect to its critical points.

**Remark 9 (SGD has an information bottleneck).** The functional (11) is equivalent to the information bottleneck principle in representation learning (Tishby et al., 1999). Minimizing this functional, explicitly, has been shown to lead to invariant representations (Achille and Soatto, 2017). Theorem 5 shows that SGD implicitly contains this bottleneck and therefore begets these properties, naturally.

**Remark 10 (ELBO prior conflicts with SGD).** Working with ELBO in practice involves one or multiple steps of SGD to minimize the energetic term along with an estimate of the KL-divergence

term, often using a factored Gaussian prior (Kingma and Welling, 2013; Jordan et al., 1999). As Theorem 5 shows, such an approach also enforces a uniform prior whose strength is determined by $\beta^{-1}$ and conflicts with the externally imposed Gaussian prior. This conflict—which fundamentally arises from using SGD to minimize the energetic term—has resulted in researchers artificially modulating the strength of the KL-divergence term using a scalar pre-factor (Mandt et al., 2016).

### 3.3 Practical implications

We will show in Section 5 that the potential $\Phi(x)$ does not depend on the optimization process, it is only a function of the dataset and the architecture. The effect of two important parameters, the learning rate $\eta$ and the mini-batch size $b$ therefore completely determines the strength of the entropic regularization term. If $\beta^{-1} \to 0$, the implicit regularization of SGD goes to zero. This implies that

$$\beta^{-1} = \frac{\eta}{2b} \text{ should not be small}$$

is a good tenet for regularization of SGD.

**Remark 11 (Learning rate should scale linearly with batch-size to generalize well).** In order to maintain the entropic regularization, the learning rate $\eta$ needs to scale linearly with the batch-size $b$. This prediction, based on Theorem 5, fits very well with empirical evidence wherein one obtains good generalization performance only with small mini-batches in deep networks (Keskar et al., 2016), or via such linear scaling (Goyal et al., 2017).

**Remark 12 (Sampling with replacement is better than without replacement).** The diffusion matrix for the case when mini-batches are sampled with replacement is very close to (2), see Appendix A.2. However, the corresponding inverse temperature is

$$\beta'^{-1} = \frac{\eta}{2b} \left( 1 - \frac{b}{N} \right) \text{ should not be small.}$$

The extra factor of $\left( 1 - \frac{b}{N} \right)$ reduces the entropic regularization in (11), as $b \to N$, the inverse temperature $\beta' \to \infty$. As a consequence, for the same learning rate $\eta$ and batch-size $b$, Theorem 5 predicts that sampling with replacement has better regularization than sampling without replacement. This effect is particularly pronounced at large batch-sizes.

## 4 Empirical characterization of SGD dynamics

Section 4.1 shows that the diffusion matrix $D(x)$ for modern deep networks is highly non-isotropic with a very low rank. We also analyze trajectories of SGD and detect periodic components using a frequency analysis in Section 4.2; this validates the prediction of Lemma 7.

We consider three networks for these experiments: a convolutional network called small-lenet, a two-layer fully-connected network on MNIST (LeCun et al., 1998) and a smaller version of the All-CNN-C architecture of Springenberg et al. (2014) on the CIFAR-10 and CIFAR-100 datasets (Krizhevsky, 2009); see Appendix E for more details.

### 4.1 Highly non-isotropic $D(x)$ for deep networks

Figs. 1 and 2 show the eigenspectrum[1] of the diffusion matrix. In all cases, it has a large fraction of almost-zero eigenvalues with a very small rank that ranges between 0.3% - 2%. Moreover, non-zero eigenvalues are spread across a vast range with a large variance.

**Remark 13 (Noise in SGD is largely independent of the weights).** The variance of noise in (3) is

$$\frac{\eta \, D(x_k)}{b} = 2 \, \beta^{-1} D(x_k).$$

---

[1] thresholded at $\lambda_{\max} \times d \times$ machine-precision. This formula is widely used, for instance, in numpy.

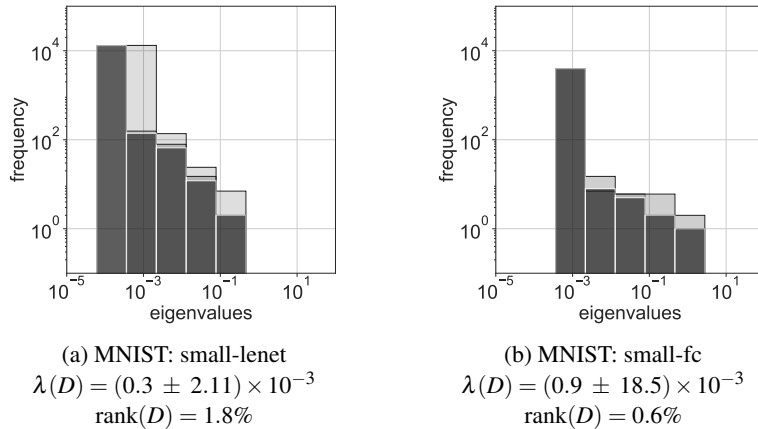

(a) MNIST: small-lenet
$\lambda(D) = (0.3 \pm 2.11) \times 10^{-3}$
$\mathrm{rank}(D) = 1.8\%$

(b) MNIST: small-fc
$\lambda(D) = (0.9 \pm 18.5) \times 10^{-3}$
$\mathrm{rank}(D) = 0.6\%$

Figure 1: Eigenspectrum of $D(x)$ at three instants during training (20%, 40% and 100% completion, darker is later). The eigenspectrum in Fig. 1b for the fully-connected network has a much smaller rank and much larger variance than the one in Fig. 1a which also performs better on MNIST. This indicates that convolutional networks are better conditioned than fully-connected networks in terms of $D(x)$.

We have plotted the eigenspectra of the diffusion matrix in Fig. 1 and Fig. 2 at three different instants, 20%, 40% and 100% training completion; they are almost indistinguishable. This implies that the variance of the mini-batch gradients in deep networks can be considered a constant, highly non-isotropic matrix.

**Remark 14 (More non-isotropic diffusion if data is diverse).** The eigenspectra in Fig. 2 for CIFAR-10 and CIFAR-100 have much larger eigenvalues and standard-deviation than those in Fig. 1, this is expected because the images in the CIFAR datasets have more variety than those in MNIST. Similarly, while CIFAR-100 has qualitatively similar images as CIFAR-10, it has $10\times$ more classes and as a result, it is a much harder dataset. This correlates well with the fact that both the mean and standard-deviation of the eigenvalues in Fig. 2b are much higher than those in Fig. 2a. Input augmentation increases the diversity of mini-batch gradients. This is seen in Fig. 2c where the standard-deviation of the eigenvalues is much higher as compared to Fig. 2a.

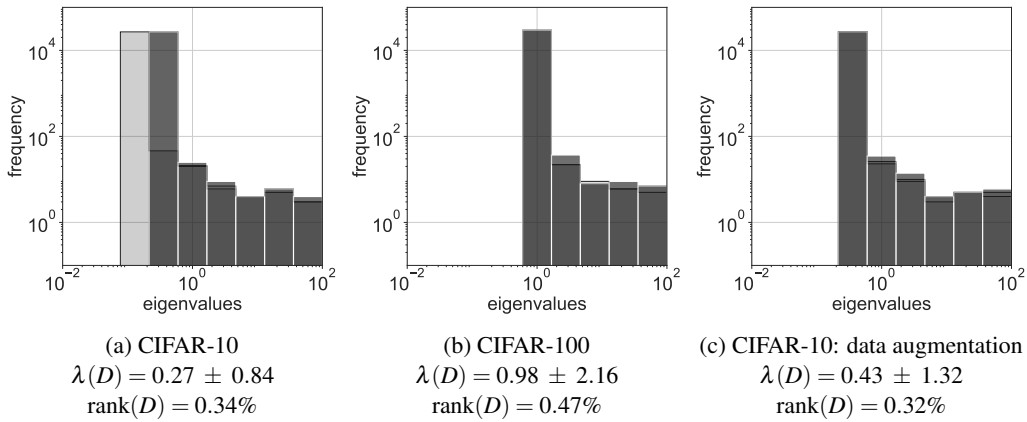

(a) CIFAR-10
$\lambda(D) = 0.27 \pm 0.84$
$\mathrm{rank}(D) = 0.34\%$

(b) CIFAR-100
$\lambda(D) = 0.98 \pm 2.16$
$\mathrm{rank}(D) = 0.47\%$

(c) CIFAR-10: data augmentation
$\lambda(D) = 0.43 \pm 1.32$
$\mathrm{rank}(D) = 0.32\%$

Figure 2: Eigenspectrum of $D(x)$ at three instants during training (20%, 40% and 100% completion, darker is later). The eigenvalues are much larger in magnitude here than those of MNIST in Fig. 1, this suggests a larger gradient diversity for CIFAR-10 and CIFAR-100. The diffusion matrix for CIFAR-100 in Fig. 2b has larger eigenvalues and is more non-isotropic and has a much larger rank than that of Fig. 2a; this suggests that gradient diversity increases with the number of classes. As Fig. 2a and Fig. 2c show, augmenting input data increases both the mean and the variance of the eigenvalues while keeping the rank almost constant.

**Remark 15 (Inverse temperature scales with the mean of the eigenspectrum).** Remark 14 shows that the mean of the eigenspectrum is large if the dataset is diverse. Based on this, we propose that

the inverse temperature $\beta$ should scale linearly with the mean of the eigenvalues of $D$:

$$\left(\frac{\eta}{b}\right)\left(\frac{1}{d}\sum_{k=1}^{d}\lambda(D)\right) = \text{constant};\qquad(14)$$

where $d$ is the number of weights. This keeps the noise in SGD constant in magnitude for different values of the learning rate $\eta$, mini-batch size $b$, architectures, and datasets. Note that other hyper-parameters which affect stochasticity such as dropout probability are implicit inside $D$.

**Remark 16 (Variance of the eigenspectrum informs architecture search).** Compare the eigen-spectra in Figs. 1a and 1b with those in Figs. 2a and 2c. The former pair shows that small-lenet which is a much better network than small-fc also has a much larger rank, i.e., the number of non-zero eigenvalues ($D(x)$ is symmetric). The second pair shows that for the same dataset, data-augmentation creates a larger variance in the eigenspectrum. This suggests that both the quantities, viz., rank of the diffusion matrix and the variance of the eigenspectrum, inform the performance of a given architecture on the dataset. Note that as discussed in Remark 15, the mean of the eigenvalues can be controlled using the learning rate $\eta$ and the batch-size $b$.

This observation is useful for automated architecture search where we can use the quantity

$$\frac{\text{rank}(D)}{d} + \text{var}(\lambda(D))$$

to estimate the efficacy of a given architecture, possibly, without even training, since $D$ does not depend on the weights much. This task currently requires enormous amounts of computational power (Zoph and Le, 2016; Baker et al., 2016; Brock et al., 2017).

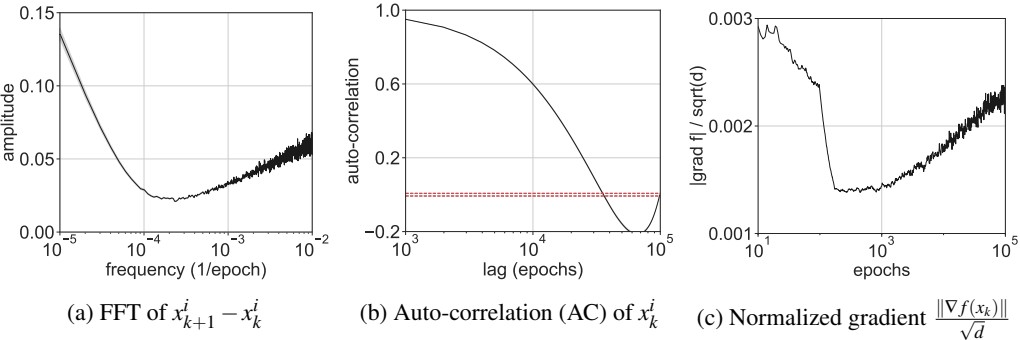

(a) FFT of $x_{k+1}^{i} - x_{k}^{i}$    (b) Auto-correlation (AC) of $x_{k}^{i}$    (c) Normalized gradient $\frac{\|\nabla f(x_k)\|}{\sqrt{d}}$

Figure 3: Fig. 3a shows the Fast Fourier Transform (FFT) of $x_{k+1}^{i} - x_{k}^{i}$ where $k$ is the number of epochs and $i$ denotes the index of the weight. Fig. 3b shows the auto-correlation of $x_{k}^{i}$ with 99% confidence bands denoted by the dotted red lines. Both Figs. 3a and 3b show the mean and one standard-deviation over the weight index $i$; the standard deviation is very small which indicates that all the weights have a very similar frequency spectrum. Figs. 3a and 3b should be compared with the FFT of white noise which should be flat and the auto-correlation of Brownian motion which quickly decays to zero, respectively. Figs. 3 and 3a therefore show that trajectories of SGD are not simply Brownian motion. Moreover the gradient at these locations is quite large (Fig. 3c).

## 4.2    ANALYSIS OF LONG-TERM TRAJECTORIES

We train a smaller version of small-fc on $7 \times 7$ down-sampled MNIST images for $10^5$ epochs and store snapshots of the weights after each epoch to get a long trajectory in the weight space. We discard the first $10^3$ epochs of training ("burnin") to ensure that SGD has reached the steady-state. The learning rate is fixed to $10^{-3}$ after this, up to $10^5$ epochs.

**Remark 17 (Low-frequency periodic components in SGD trajectories).** Iterates of SGD, after it reaches the neighborhood of a critical point $\|\nabla f(x_k)\| \le \varepsilon$, are expected to perform Brownian motion with variance $\text{var}(\nabla f_b(x))$, the FFT in Fig. 3a would be flat if this were so. Instead, we see low-frequency modes in the trajectory that are indicators of a periodic dynamics of the force $j(x)$. These modes are not sharp peaks in the FFT because $j(x)$ can be a non-linear function of the

weights thereby causing the modes to spread into all dimensions of $x$. The FFT is dominated by jittery high-frequency modes on the right with a slight increasing trend; this suggests the presence of colored noise in SGD at high-frequencies.

The auto-correlation (AC) in Fig. 3b should be compared with the AC for Brownian motion which decays to zero very quickly and stays within the red confidence bands (99%). Our iterates are significantly correlated with each other even at very large lags. This further indicates that trajectories of SGD do not perform Brownian motion.

**Remark 18 (Gradient magnitude in deep networks is always large).** Fig. 3c shows that the full-gradient computed over the entire dataset (without burnin) does not decrease much with respect to the number of epochs. While it is expected to have a non-zero gradient norm because SGD only converges to a neighborhood of a critical point for non-zero learning rates, the magnitude of this gradient norm is quite large. This magnitude drops only by about a factor of 3 over the next $10^5$ epochs. The presence of a non-zero $j(x)$ also explains this, it causes SGD to be away from critical points, this phenomenon is made precise in Theorem 22. Let us note that a similar plot is also seen in Shwartz-Ziv and Tishby (2017) for the per-layer gradient magnitude.

## 5 SGD FOR DEEP NETWORKS IS OUT-OF-EQUILIBRIUM

This section now gives an explicit formula for the potential $\Phi(x)$. We also discuss implications of this for generalization in Section 5.3.

The fundamental difficulty in obtaining an explicit expression for $\Phi$ is that even if the diffusion matrix $D(x)$ is full-rank, there need not exist a function $\Phi(x)$ such that $\nabla\Phi(x) = D^{-1}(x)\,\nabla f(x)$ at all $x \in \Omega$. We therefore split the analysis into two cases:

(i) a local analysis near any critical point $\nabla f(x) = 0$ where we linearize $\nabla f(x) = Fx$ and $\nabla\Phi(x) = Ux$ to compute $U = G^{-1}\,F$ for some $G$, and

(ii) the general case where $\nabla\Phi(x)$ cannot be written as a local rotation and scaling of $\nabla f(x)$.

Let us introduce these cases with an example from Noh and Lee (2015).

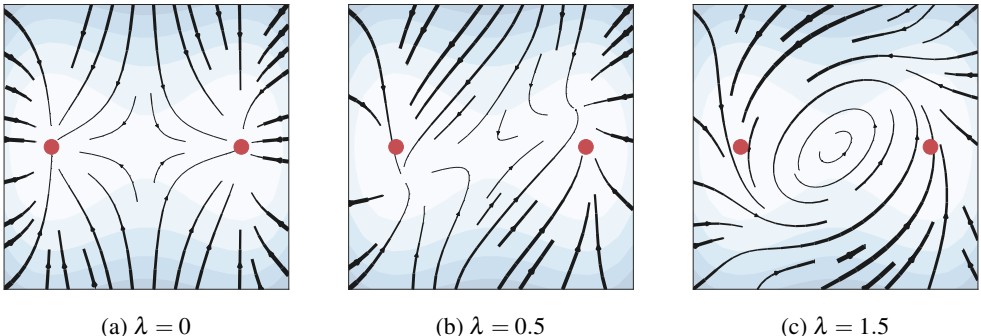

(a) $\lambda = 0$        (b) $\lambda = 0.5$        (c) $\lambda = 1.5$

Figure 4: Gradient field for the dynamics in Example 19: line-width is proportional to the magnitude of the gradient $\|\nabla f(x)\|$, red dots denote the most likely locations of the steady-state $e^{-\Phi}$ while the potential $\Phi$ is plotted as a contour map. The critical points of $f(x)$ and $\Phi(x)$ are the same in Fig. 4a, namely $(\pm 1, 0)$, because the force $j(x) = 0$. For $\lambda = 0.5$ in Fig. 4b, locations where $\nabla f(x) = 0$ have shifted slightly as predicted by Theorem 22. The force field also has a distinctive rotation component, see Remark 21. In Fig. 4c with a large $\|j(x)\|$, SGD converges to limit cycles around the saddle point at the origin. This is highly surprising and demonstrates that the solutions obtained by SGD may be very different from local minima.

**Example 19 (Double-well potential with limit cycles).** Define

$$\Phi(x) = \frac{(x_1^2 - 1)^2}{4} + \frac{x_2^2}{2}.$$

Instead of constructing a diffusion matrix $D(x)$, we will directly construct different gradients $\nabla f(x)$ that lead to the same potential $\Phi$; these are equivalent but the later is much easier. The dynamics is

given by $dx = -\nabla f(x) \, dt + \sqrt{2} \, dW(t)$, where $\nabla f(x) = -j(x) + \nabla \Phi(x)$. We pick $j = \lambda e^{\Phi} J^{\text{ss}}(x)$ for some parameter $\lambda > 0$ where

$$J^{\text{ss}}(x) = e^{-\frac{(x_1^2 + x_2^2)^2}{4}} (-x_2, \, x_1).$$

Note that this satisfies (6) and does not change $\rho^{\text{ss}} = e^{-\Phi}$. Fig. 4 shows the gradient field $f(x)$ along with a discussion.

## 5.1 LINEARIZATION AROUND A CRITICAL POINT

Without loss of generality, let $x = 0$ be a critical point of $f(x)$. This critical point can be a local minimum, maximum, or even a saddle point. We linearize the gradient around the origin and define a fixed matrix $F \in \mathbb{R}^{d \times d}$ (the Hessian) to be $\nabla f(x) = Fx$. Let $D = D(0)$ be the constant diffusion matrix matrix. The dynamics in (3) can now be written as

$$dx = -Fx \, dt + \sqrt{2\beta^{-1} D} \, dW(t). \tag{15}$$

**Lemma 20 (Linearization).** *The matrix $F$ in (15) can be uniquely decomposed into*

$$F = (D + Q) \, U; \tag{16}$$

*$D$ and $Q$ are the symmetric and anti-symmetric parts of a matrix $G$ with $GF^\top - FG^\top = 0$, to get $\Phi(x) = \frac{1}{2} x^\top U x$.*

The above lemma is a classical result if the critical point is a local minimum, i.e., if the loss is locally convex near $x = 0$; this case has also been explored in machine learning before (Mandt et al., 2016). We refer to Kwon et al. (2005) for the proof that linearizes around any critical point.

**Remark 21 (Rotation of gradients).** We see from Lemma 20 that, near a critical point,

$$\nabla f = (D + Q) \, \nabla \Phi - \beta^{-1} \nabla \cdot D - \beta^{-1} \nabla \cdot Q \tag{17}$$

up to the first order. This suggests that the effect of $j(x)$ is to rotate the gradient field and move the critical points, also seen in Fig. 4b. Note that $\nabla \cdot D = 0$ and $\nabla \cdot Q = 0$ in the linearized analysis.

## 5.2 GENERAL CASE

We next give the general expression for the deviation of the critical points $\nabla \Phi$ from those of the original loss $\nabla f$.

**A-type stochastic integration:** A Fokker-Planck equation is a deterministic partial differential equation (PDE) and every steady-state distribution, $\rho^{\text{ss}} \propto e^{-\beta \Phi}$ in this case, has a unique such PDE that achieves it. However, the same PDE can be tied to different SDEs depending on the stochastic integration scheme, e.g., Ito, Stratonovich (Risken, 1996; Oksendal, 2003), Hanggi (Hänggi, 1978), $\alpha$-type etc. An "A-type" interpretation is one such scheme (Ao et al., 2007; Shi et al., 2012). It is widely used in non-equilibrium studies in physics and biology (Wang et al., 2008; Zhu et al., 2004) because it allows one to compute the steady-state distribution easily; its implications are supported by other mathematical analyses such as Tel et al. (1989); Qian (2014).

The main result of the section now follows. It exploits the A-type interpretation to compute the difference between the most likely locations of SGD which are given by the critical points of the potential $\Phi(x)$ and those of the original loss $f(x)$.

**Theorem 22 (Most likely locations are not the critical points of the loss).** *The Ito SDE*

$$dx = -\nabla f(x) \, dt + \sqrt{2\beta^{-1} D(x)} \, dW(t)$$

*is equivalent to the A-type SDE (Ao et al., 2007; Shi et al., 2012)*

$$dx = -\left( D(x) + Q(x) \right) \nabla \Phi(x) \, dt + \sqrt{2\beta^{-1} D(x)} \, dW(t) \tag{18}$$

*with the same steady-state distribution $\rho^{ss} \propto e^{-\beta\Phi(x)}$ and Fokker-Planck equation (FP) if*

$$\nabla f(x) = \left(D(x) + Q(x)\right) \nabla\Phi(x) - \beta^{-1}\nabla \cdot \left(D(x) + Q(x)\right). \quad (19)$$

*The anti-symmetric matrix $Q(x)$ and the potential $\Phi(x)$ can be explicitly computed in terms of the gradient $\nabla f(x)$ and the diffusion matrix $D(x)$. The potential $\Phi(x)$ does not depend on $\beta$.*

See Appendix F.4 for the proof. It exploits the fact that the the Ito SDE (3) and the A-type SDE (18) should have the same Fokker-Planck equations because they have the same steady-state distributions.

**Remark 23 (SGD is far away from critical points).** The time spent by a Markov chain at a state $x$ is proportional to its steady-state distribution $\rho^{ss}(x)$. While it is easily seen that SGD does not converge in the Cauchy sense due to the stochasticity, it is very surprising that it may spend a significant amount of time away from the critical points of the original loss. If $D(x) + Q(x)$ has a large divergence, the set of states with $\nabla\Phi(x) = 0$ might be drastically different than those with $\nabla f(x) = 0$. This is also seen in example Fig. 4c; in fact, SGD may even converge around a saddle point.

This also closes the logical loop we began in Section 3 where we assumed the existence of $\rho^{ss}$ and defined the potential $\Phi$ using it. Lemma 20 and Theorem 22 show that both can be defined uniquely in terms of the original quantities, *i.e.*, the gradient term $\nabla f(x)$ and the diffusion matrix $D(x)$. There is no ambiguity as to whether the potential $\Phi(x)$ results in the steady-state $\rho^{ss}(x)$ or vice-versa.

**Remark 24 (Consistent with the linear case).** Theorem 22 presents a picture that is completely consistent with Lemma 20. If $j(x) = 0$ and $Q(x) = 0$, or if $Q$ is a constant like the linear case in Lemma 20, the divergence of $Q(x)$ in (19) is zero.

**Remark 25 (Out-of-equilibrium effect can be large even if $D$ is constant).** The presence of a $Q(x)$ with non-zero divergence is the consequence of a non-isotropic $D(x)$ and it persists even if $D$ is constant and independent of weights $x$. So long as $D$ is not isotropic, as we discussed in the beginning of Section 5, there need not exist a function $\Phi(x)$ such that $\nabla\Phi(x) = D^{-1} \nabla f(x)$ at all $x$. This is also seen in our experiments, the diffusion matrix is almost constant with respect to weights for deep networks, but consequences of out-of-equilibrium behavior are still seen in Section 4.2.

**Remark 26 (Out-of-equilibrium effect increases with $\beta^{-1}$).** The effect predicted by (19) becomes more pronounced if $\beta^{-1} = \frac{\eta}{2b}$ is large. In other words, small batch-sizes or high learning rates cause SGD to be drastically out-of-equilibrium. Theorem 5 also shows that as $\beta^{-1} \to 0$, the implicit entropic regularization in SGD vanishes. Observe that these are exactly the conditions under which we typically obtain good generalization performance for deep networks (Keskar et al., 2016; Goyal et al., 2017). This suggests that non-equilibrium behavior in SGD is crucial to obtain good generalization performance, especially for high-dimensional models such as deep networks where such effects are expected to be more pronounced.

## 5.3 GENERALIZATION

It was found that solutions of discrete learning problems that generalize well belong to dense clusters in the weight space (Baldassi et al., 2015; 2016). Such dense clusters are exponentially fewer compared to isolated solutions. To exploit these observations, the authors proposed a loss called "local entropy" that is out-of-equilibrium by construction and can find these well-generalizable solutions easily. This idea has also been successful in deep learning where Chaudhari et al. (2016) modified SGD to seek solutions in "wide minima" with low curvature to obtain improvements in generalization performance as well as convergence rate (Chaudhari et al., 2017a).

Local entropy is a smoothed version of the original loss given by

$$f_\gamma(x) = -\log\left(G_\gamma * e^{-f(x)}\right),$$

where $G_\gamma$ is a Gaussian kernel of variance $\gamma$. Even with an isotropic diffusion matrix, the steady-state distribution with $f_\gamma(x)$ as the loss function is $\rho_\gamma^{ss}(x) \propto e^{-\beta f_\gamma(x)}$. For large values of $\gamma$, the new loss makes the original local minima exponentially less likely. In other words, local entropy does not rely on non-isotropic gradient noise to obtain out-of-equilibrium behavior, it gets it explicitly, by

construction. This is also seen in Fig. 4c: if SGD is drastically out-of-equilibrium, it converges around the "wide" saddle point region at the origin which has a small local entropy.

Actively constructing out-of-equilibrium behavior leads to good generalization in practice. Our evidence that SGD on deep networks itself possesses out-of-equilibrium behavior then indicates that SGD for deep networks generalizes well because of such behavior.

## 6  RELATED WORK

**SGD, variational inference and implicit regularization:**    The idea that SGD is related to variational inference has been seen in machine learning before (Duvenaud et al., 2016; Mandt et al., 2016) under assumptions such as quadratic steady-states; for instance, see Mandt et al. (2017) for methods to approximate steady-states using SGD. Our results here are very different, we would instead like to understand properties of SGD itself. Indeed, in full generality, SGD performs variational inference using a new potential $\Phi$ that it implicitly constructs given an architecture and a dataset.

It is widely believed that SGD is an implicit regularizer, see Zhang et al. (2016); Neyshabur et al. (2017); Shwartz-Ziv and Tishby (2017) among others. This belief stems from its remarkable empirical performance. Our results show that such intuition is very well-placed. Thanks to the special architecture of deep networks where gradient noise is highly non-isotropic, SGD helps itself to a potential $\Phi$ with properties that lead to both generalization and acceleration.

**SGD and noise:**    Noise is often added in SGD to improve its behavior around saddle points for non-convex losses, see Lee et al. (2016); Anandkumar and Ge (2016); Ge et al. (2015). It is also quite indispensable for training deep networks (Hinton and Van Camp, 1993; Srivastava et al., 2014; Kingma et al., 2015; Gulcehre et al., 2016; Achille and Soatto, 2017). There is however a disconnect between these two directions due to the fact that while adding external gradient noise helps in theory, it works poorly in practice (Neelakantan et al., 2015; Chaudhari and Soatto, 2015). Instead, "noise tied to the architecture" works better, e.g., dropout, or small mini-batches. Our results close this gap and show that SGD crucially leverages the highly degenerate noise induced by the architecture.

**Gradient diversity:**    Yin et al. (2017) construct a scalar measure of the gradient diversity given by $\sum_k \|\nabla f_k(x)\|/\|\nabla f(x)\|$, and analyze its effect on the maximum allowed batch-size in the context of distributed optimization.

**Markov Chain Monte Carlo:**    MCMC methods that sample from a negative log-likelihood $\Phi(x)$ have employed the idea of designing a force $j = \nabla\Phi - \nabla f$ to accelerate convergence, see Ma et al. (2015) for a thorough survey, or Pavliotis (2016); Kaiser et al. (2017) for a rigorous treatment. We instead compute the potential $\Phi$ given $\nabla f$ and $D$, which necessitates the use of techniques from physics. In fact, our results show that since $j \neq 0$ for deep networks due to non-isotropic gradient noise, very simple algorithms such as SGLD by Welling and Teh (2011) also benefit from the acceleration that their sophisticated counterparts aim for (Ding et al., 2014; Chen et al., 2016).

## 7  DISCUSSION

The continuous-time point-of-view used in this paper gives access to general principles that govern SGD, such analyses are increasingly becoming popular (Wibisono et al., 2016; Chaudhari et al., 2017b). However, in practice, deep networks are trained for only a few epochs with discrete-time updates. Closing this gap is an important future direction. A promising avenue towards this is that for typical conditions in practice such as small mini-batches or large learning rates, SGD converges to the steady-state distribution quickly (Raginsky et al., 2017).

## 8 ACKNOWLEDGMENTS

PC would like to thank Adam Oberman for introducing him to the JKO functional. The authors would also like to thank Alhussein Fawzi for numerous discussions during the conception of this paper and his contribution to its improvement. This research was supported by ARO W911NF-17-1-0304, ONR N00014-17-1-2072, AFOSR FA9550-15-1-0229.

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

## A  DIFFUSION MATRIX $D(x)$

In this section we denote $g_k := \nabla f_k(x)$ and $g := \nabla f(x) = \frac{1}{N} \sum_{k=1}^{N} g_k$. Although we drop the dependence of $g_k$ on $x$ to keep the notation clear, we emphasize that the diffusion matrix $D$ depends on the weights $x$.

### A.1  WITH REPLACEMENT

Let $i_1, \ldots, i_b$ be $b$ iid random variables in $\{1, 2, \ldots, N\}$. We would like to compute

$$\text{var}\left(\frac{1}{b} \sum_{j=1}^{b} g_{i_j}\right) = \mathbb{E}_{i_1,\ldots,i_b} \left\{ \left(\frac{1}{b} \sum_{j=1}^{b} g_{i_j} - g\right) \left(\frac{1}{b} \sum_{j=1}^{b} g_{i_j} - g\right)^\top \right\}.$$

Note that we have that for any $j \neq k$, the random vectors $g_{i_j}$ and $g_{i_k}$ are independent. We therefore have

$$\text{covar}(g_{i_j}, g_{i_k}) = 0 = \mathbb{E}_{i_j, i_k} \left\{ (g_{i_j} - g)(g_{i_k} - g)^\top \right\}$$

We use this to obtain

$$\text{var}\left(\frac{1}{b} \sum_{j=1}^{b} g_{i_j}\right) = \frac{1}{b^2} \sum_{j=1}^{b} \text{var}(g_{i_j}) = \frac{1}{Nb} \sum_{k=1}^{N} \left((g_k - g)(g_k - g)^\top\right)$$

$$= \frac{1}{b} \left( \frac{\sum_{k=1}^{N} g_k g_k^\top}{N} - g\,g^\top \right).$$

We will set

$$D(x) = \frac{1}{N} \left( \sum_{k=1}^{N} g_k g_k^\top \right) - g\,g^\top. \tag{A1}$$

and assimilate the factor of $b^{-1}$ in the inverse temperature $\beta$.

### A.2  WITHOUT REPLACEMENT

Let us define an indicator random variable $\mathbf{1}_{i \in b}$ that denotes if an example $i$ was sampled in batch $b$. We can show that

$$\text{var}(\mathbf{1}_{i \in b}) = \frac{b}{N} - \frac{b^2}{N^2},$$

and for $i \neq j$,

$$\text{covar}(\mathbf{1}_{i \in b}, \mathbf{1}_{j \in b}) = -\frac{b(N - b)}{N^2(N - 1)}.$$

Similar to Li et al. (2017a), we can now compute

$$\text{var}\left(\frac{1}{b} \sum_{k=1}^{N} g_k \mathbf{1}_{k \in b}\right) = \frac{1}{b^2} \text{var}\left(\sum_{k=1}^{N} g_k \mathbf{1}_{k \in b}\right)$$

$$= \frac{1}{b^2} \sum_{k=1}^{N} g_k g_k^\top \text{var}(\mathbf{1}_{k \in b}) + \frac{1}{b^2} \sum_{i,j=1,\ i \neq j}^{N} g_i g_j^\top \text{covar}(\mathbf{1}_{i \in b}, \mathbf{1}_{j \in b})$$

$$= \frac{1}{b} \left(1 - \frac{b}{N}\right) \left[ \frac{\sum_{k=1}^{N} g_k g_k^\top}{N - 1} - \left(1 - \frac{1}{N-1}\right) g\,g^\top \right].$$

We will again set

$$D(x) = \frac{1}{N-1} \left( \sum_{k=1}^{N} g_k g_k^\top \right) - \left(1 - \frac{1}{N-1}\right) g\,g^\top \tag{A2}$$

and assimilate the factor of $b^{-1}\left(1 - \frac{b}{N}\right)$ that depends on the batch-size in the inverse temperature $\beta$.

## B    DISCUSSION ON ASSUMPTION 4

Let $F(\rho)$ be as defined in (11). In non-equilibrium thermodynamics, it is assumed that the local entropy production is a product of the force $-\nabla\left(\frac{\delta F}{\delta\rho}\right)$ from (A8) and the probability current $-J(x,t)$ from (FP). This assumption in this form was first introduced by Prigogine (1955) based on the works of Onsager (1931a;b). See Frank (2005, Sec. 4.5) for a mathematical treatment and Jaynes (1980) for further discussion. The rate of entropy $(S_i)$ increase is given by

$$\beta^{-1}\frac{dS_i}{dt} = \int_{x\in\Omega}\nabla\left(\frac{\delta F}{\delta\rho}\right)J(x,t)\,dx.$$

This can now be written using (A8) again as

$$\beta^{-1}\frac{dS_i}{dt} = \int \rho\,D:\left(\nabla\frac{\delta F}{\delta\rho}\right)\left(\nabla\frac{\delta F}{\delta\rho}\right)^\top + \int j\rho\left(\nabla\frac{\delta F}{\delta\rho}\right)\,dx.$$

The first term in the above expression is non-negative, in order to ensure that $\frac{dS_i}{dt}\geq 0$, we require

$$0 = \int j\rho\left(\nabla\frac{\delta F}{\delta\rho}\right)\,dx$$
$$= \int \nabla\cdot(j\rho)\left(\frac{\delta F}{\delta\rho}\right)\,dx;$$

where the second equality again follows by integration by parts. It can be shown (Frank, 2005, Sec. 4.5.5) that the condition in Assumption 4, viz., $\nabla\cdot j(x) = 0$, is sufficient to make the above integral vanish and therefore for the entropy generation to be non-negative.

## C    SOME PROPERTIES OF THE FORCE $j$

The Fokker-Planck equation (FP) can be written in terms of the probability current as

$$0 = \rho_t^{ss} = \nabla\cdot\left(-j\,\rho^{ss} + D\,\nabla\Phi\,\rho^{ss} - \beta^{-1}(\nabla\cdot D)\,\rho^{ss} + \beta^{-1}\nabla\cdot(D\rho^{ss})\right)$$
$$= \nabla\cdot J^{ss}.$$

Since we have $\rho^{ss} \propto e^{-\beta\Phi(x)}$, from the observation (7), we also have that

$$0 = \rho_t^{ss} = \nabla\cdot\left(D\,\nabla\Phi\,\rho^{ss} + \beta^{-1}D\,\nabla\rho^{ss}\right),$$

and consequently,

$$0 = \nabla\cdot(j\,\rho^{ss})$$
$$\Rightarrow \quad j(x) = \frac{J^{ss}}{\rho^{ss}}. \tag{A3}$$

In other words, the conservative force is non-zero only if detailed balance is broken, i.e., $J^{ss}\neq 0$. We also have

$$0 = \nabla\cdot(j\,\rho^{ss})$$
$$= \rho^{ss}\left(\nabla\cdot j - j\cdot\nabla\Phi\right),$$

which shows using Assumption 4 and $\rho^{ss}(x) > 0$ for all $x\in\Omega$ that $j(x)$ is always orthogonal to the gradient of the potential

$$0 = j(x)\cdot\nabla\Phi(x)$$
$$= j(x)\cdot\nabla\rho^{ss}. \tag{A4}$$

Using the definition of $j(x)$ in (8), we have detailed balance when

$$\nabla f(x) = D(x)\,\nabla\Phi(x) - \beta^{-1}\nabla\cdot D(x). \tag{A5}$$

## D  HEAT EQUATION AS A GRADIENT FLOW

As first discovered in the works of Jordan, Kinderleherer and Otto (Jordan et al., 1998; Otto, 2001), certain partial differential equations can be seen as coming from a variational principle, i.e., they perform steepest descent with respect to functionals of their state distribution. Section 3 is a generalization of this idea, we give a short overview here with the heat equation. The heat equation

$$\rho_t = \nabla \cdot (\nabla \rho),$$

can be written as the steepest descent for the Dirichlet energy functional

$$\frac{1}{2} \int_\Omega |\nabla \rho|^2 \, dx.$$

However, the same PDE can also be seen as the gradient flow of the negative Shannon entropy in the Wasserstein metric (Santambrogio, 2017; 2015),

$$-H(\rho) = \int_\Omega \rho(x) \log \rho(x) \, dx.$$

More precisely, the sequence of iterated minimization problems

$$\rho_{k+1}^\tau \in \arg \min_\rho \left\{ -H(\rho) + \frac{\mathbb{W}_2^2(\rho, \rho_k^\tau)}{2\tau} \right\} \tag{A6}$$

converges to trajectories of the heat equation as $\tau \to 0$. This equivalence is extremely powerful because it allows us to interpret, and modify, the functional $-H(\rho)$ that PDEs such as the heat equation implicitly minimize.

This equivalence is also quite natural, the heat equation describes the probability density of pure Brownian motion: $dx = \sqrt{2} \, dW(t)$. The Wasserstein point-of-view suggests that Brownian motion maximizes the entropy of its state distribution, while the Dirichlet functional suggests that it minimizes the total-variation of its density. These are equivalent. While the latter has been used extensively in image processing, our paper suggests that the entropic regularization point-of-view is very useful to understand SGD for machine learning.

## E  EXPERIMENTAL SETUP

We consider the following three networks on the MNIST (LeCun et al., 1998) and the CIFAR-10 and CIFAR-100 datasets (Krizhevsky, 2009).

(i) **small-lenet:** a smaller version of LeNet (LeCun et al., 1998) on MNIST with batch-normalization and dropout (0.1) after both convolutional layers of 8 and 16 output channels, respectively. The fully-connected layer has 128 hidden units. This network has $13,338$ weights and reaches about 0.75% training and validation error.

(ii) **small-fc:** a fully-connected network with two-layers, batch-normalization and rectified linear units that takes $7 \times 7$ down-sampled images of MNIST as input and has 64 hidden units. Experiments in Section 4.2 use a smaller version of this network with 16 hidden units and 5 output classes ($30,000$ input images); this is called **tiny-fc**.

(iii) **small-allcnn:** this a smaller version of the fully-convolutional network for CIFAR-10 and CIFAR-100 introduced by Springenberg et al. (2014) with batch-normalization and $12, 24$ output channels in the first and second block respectively. It has $26,982$ weights and reaches about 11% and 17% training and validation errors, respectively.

We train the above networks with SGD with appropriate learning rate annealing and Nesterov's momentum set to 0.9. We do not use any data-augmentation and pre-process data using global contrast normalization with ZCA for CIFAR-10 and CIFAR-100.

We use networks with about $20,000$ weights to keep the eigen-decomposition of $D(x) \in \mathbb{R}^{d \times d}$ tractable. These networks however possess all the architectural intricacies such as convolutions, dropout, batch-normalization etc. We evaluate $D(x)$ using (2) with the network in evaluation mode.

# F  PROOFS

## F.1  THEOREM 5

The KL-divergence is non-negative: $F(\rho) \geq 0$ with equality if and only if $\rho = \rho^{\text{ss}}$. The expression in (11) follows after writing

$$\log \rho^{\text{ss}} = -\beta \Phi - \log Z(\beta).$$

We now show that $\frac{dF(\rho)}{dt} \leq 0$ with equality only at $\rho = \rho^{\text{ss}}$ when $F(\rho)$ reaches its minimum and the Fokker-Planck equation achieves its steady-state. The first variation (Santambrogio, 2015) of $F(\rho)$ computed from (11) is

$$\frac{\delta F}{\delta \rho}(\rho) = \Phi(x) + \beta^{-1} \left( \log \rho + 1 \right), \tag{A7}$$

which helps us write the Fokker-Planck equation (FP) as

$$\rho_t = \nabla \cdot \left( -j \, \rho + \rho \, D \, \nabla \left( \frac{\delta F}{\delta \rho} \right) \right). \tag{A8}$$

Together, we can now write

$$\frac{dF(\rho)}{dt} = \int_{x \in \Omega} \rho_t \, \frac{\delta F}{\delta \rho} \, dx$$

$$= \int_{x \in \Omega} \frac{\delta F}{\delta \rho} \, \nabla \cdot (-j \, \rho) \, dx + \int_{x \in \Omega} \frac{\delta F}{\delta \rho} \, \nabla \cdot \left( \rho \, D \, \nabla \left( \frac{\delta F}{\delta \rho} \right) \right) \, dx.$$

As we show in Appendix B, the first term above is zero due to Assumption 4. Under suitable boundary condition on the Fokker-Planck equation which ensures that no probability mass flows across the boundary of the domain $\partial \Omega$, after an integration by parts, the second term can be written as

$$\frac{dF(\rho)}{dt} = -\int_{x \in \Omega} \rho \, D : \left( \nabla \frac{\delta F}{\delta \rho}(\rho) \right) \left( \nabla \frac{\delta F}{\delta \rho}(\rho) \right)^\top \, dx$$

$$\leq 0.$$

In the above expression, $A : B$ denotes the matrix dot product $A : B = \sum_{ij} A_{ij} B_{ij}$. The final inequality with the quadratic form holds because $D(x) \succeq 0$ is a covariance matrix. Moreover, we have from (A7) that

$$\frac{dF(\rho^{\text{ss}})}{dt} = 0.$$

## F.2  LEMMA 6

The forward implication can be checked by substituting $\rho^{\text{ss}}(x) \propto e^{-c \, \beta f(x)}$ in the Fokker-Planck equation (FP) while the reverse implication is true since otherwise (A4) would not hold.

## F.3  LEMMA 7

The Fokker-Planck operator written as

$$L \, \rho = \nabla \cdot \left( -j \, \rho + D \, \nabla \Phi \, \rho - \beta^{-1} \, (\nabla \cdot D) \, \rho + \beta^{-1} \nabla \cdot (D \, \rho) \right)$$

from (8) and (FP) can be split into two operators

$$L = L_S + L_A,$$

where the symmetric part is

$$L_S \rho = \nabla \cdot \left( D \, \nabla \Phi \, \rho - \beta^{-1} \, (\nabla \cdot D) \, \rho + \beta^{-1} \nabla \cdot (D \, \rho) \right) \tag{A9}$$

and the anti-symmetric part is

$$
\begin{aligned}
L_A \rho &= \nabla \cdot (-j\rho) \\
&= \nabla \cdot \left( -D \, \nabla \Phi \, \rho + \nabla f \rho + \beta^{-1} (\nabla \cdot D) \rho \right) \\
&= \nabla \cdot \left( \beta^{-1} \, D \, \nabla \rho + \nabla f \rho + \beta^{-1} (\nabla \cdot D) \rho \right).
\end{aligned}
\tag{A10}
$$

We first note that $L_A$ does not affect $F(\rho)$ in Theorem 5. For solutions of $\rho_t = L_A \, \rho$, we have

$$
\begin{aligned}
\frac{d}{dt} F(\rho) &= \int_\Omega \frac{\delta F}{\delta \rho} \, \rho_t \, dx \\
&= \int_\Omega \frac{\delta F}{\delta \rho} \, \nabla \cdot (-j \, \rho) \, dx \\
&= 0,
\end{aligned}
$$

by Assumption 4. The dynamics of the anti-symmetric operator is thus completely deterministic and conserves $F(\rho)$. In fact, the equation (A10) is known as the Liouville equation (Frank, 2005) and describes the density of a completely deterministic dynamics given by

$$\dot{x} = j(x); \tag{A11}$$

where $j(x) = J^{\text{ss}}/\rho^{\text{ss}}$ from Appendix C. On account of the trajectories of the Liouville operator being deterministic, they are also the most likely ones under the steady-state distribution $\rho^{\text{ss}} \propto e^{-\beta\Phi}$.

## F.4 THEOREM 22

All the matrices below depend on the weights $x$; we suppress this to keep the notation clear. Our original SDE is given by

$$dx = -\nabla f \, dt + \sqrt{2\beta^{-1} D} \, dW(t).$$

We will transform the original SDE into a new SDE

$$G \, dx = -\nabla \Phi \, dt + \sqrt{2\beta^{-1} S} \, dW(t) \tag{A12}$$

where $S$ and $A$ are the symmetric and anti-symmetric parts of $G^{-1}$,

$$S = \frac{G^{-1} + G^{-T}}{2},$$

$$A = G^{-1} - S.$$

Since the two SDEs above are equal to each other, both the deterministic and the stochastic terms have to match. This gives

$$\nabla f(x) = G \, \nabla \Phi(x)$$

$$D = \frac{G + G^\top}{2}$$

$$Q = \frac{G - G^\top}{2}.$$

Using the above expression, we can now give an explicit, although formal, expression for the potential:

$$\Phi(x) = \int_0^1 \left( G^{-1}(\Gamma(s)) \, \nabla f(\Gamma(s)) \right) \cdot d\Gamma(s); \tag{A13}$$

where $\Gamma : [0, 1] \to \Omega$ is any curve such that $\Gamma(1) = x$ and $\Gamma(0) = x(0)$ which is the initial condition of the dynamics in (3). Note that $\Phi(x)$ does not depend on $\beta$ because $G(x)$ does not depend on $\beta$.

We now write the modified SDE (A12) as a second-order Langevin system after introducing a velocity variable $p$ with $q \triangleq x$ and mass $m$:

$$
\begin{aligned}
dq &= \frac{p}{m} \, dt \\
dp &= -(S+A) \, \frac{p}{m} \, dt - \nabla_q \Phi(q) \, dt + \sqrt{2\beta^{-1} \, S} \, dW(t).
\end{aligned}
\tag{A14}
$$

The key idea in Yin and Ao (2006) is to compute the Fokker-Planck equation of the system above and take its zero-mass limit. The steady-state distribution of this equation, which is also known as the Klein-Kramer's equation, is

$$
\rho^{\text{ss}}(q,p) = \frac{1}{Z'(\beta)} \, \exp\left(-\beta \, \Phi(q) - \frac{\beta p^2}{2m}\right) ;
\tag{A15}
$$

where the position and momentum variables are decoupled. The zero-mass limit is given by

$$
\begin{aligned}
\rho_t &= \nabla \cdot G \left(\nabla \Phi \, \rho + \beta^{-1} \, \nabla \rho\right), \\
&= \nabla \cdot \left(D \, \nabla \Phi \, \rho + Q \, \nabla \Phi \, \rho + (D+Q) \, \beta^{-1} \, \nabla \rho\right) \\
&= \nabla \cdot \left(D \, \nabla \Phi \, \rho + Q \, \nabla \Phi \, \rho\right) + \nabla \cdot \left(D \, \beta^{-1} \, \nabla \rho\right) + \beta^{-1} \underbrace{\nabla \cdot (Q \, \nabla) \, \rho}_{*}
\end{aligned}
\tag{A16}
$$

We now exploit the fact that $Q$ is defined to be an anti-symmetric matrix. Note that $\sum_{i,j} \partial_i \partial_j (Q_{ij} \rho) = 0$ because $Q$ is anti-symmetric. Rewrite the third term on the last step $(*)$ as

$$
\begin{aligned}
\nabla \cdot (Q \, \nabla \rho) &= \sum_{ij} \partial_i \left(Q_{ij} \, \partial_j \rho\right) \\
&= -\sum_{ij} \partial_i \left(\partial_j Q_{ij}\right) \rho \\
&= -\nabla \cdot \left(\nabla \cdot Q\right) \rho.
\end{aligned}
\tag{A17}
$$

We now use the fact that (3) has $\rho^{\text{ss}} \propto e^{-\beta \Phi}$ as the steady-state distribution as well. Since the steady-state distribution is uniquely determined by a Fokker-Planck equation, the two equations (FP) and (A16) are the same. Let us decompose the second term in (FP):

$$
\begin{aligned}
&\beta^{-1} \sum_{i,j} \partial_i \partial_j \left[D_{ij}(x) \rho(x)\right] \\
&= \beta^{-1} \sum_{i,j} \partial_i \{(\partial_j D_{ij}) \, \rho\} + \beta^{-1} \sum_{i,j} \partial_i \{D_{ij} \partial_j \rho\}.
\end{aligned}
$$

Observe that the brown terms are equal. Moving the blue terms together and matching the drift terms in the two Fokker-Planck equations then gives

$$
\nabla f = (D+Q) \, \nabla \Phi - \beta^{-1} \nabla \cdot D - \beta^{-1} \nabla \cdot Q
$$

The critical points of $\Phi$ are different from those of the original loss $f$ by a term that is $\beta^{-1} \nabla \cdot (D+Q)$.

