# OpenReview forum: "Stochastic gradient descent performs variational inference, converges to limit cycles for deep networks"
_ICLR.cc/2018/Conference — Accept (Poster)_

### Official Review · AnonReviewer3 · 2017-11-26
**A variational analysis of SGD as a non-equilibrium process.**

**Rating:** 8
**Confidence:** 5

**Review:**

The paper takes a closer look at the analysis of SGD as variational inference, first proposed by Duvenaud et al. 2016
and Mandt et al. 2016. In particular, the authors point out that in general, SGD behaves quite differently from Langevin diffusion due to the multivariate nature of the Gaussian noise. As the authors show based on the Fokker-Planck equation of the underlying stochastic process, there exists a conservative current (a gradient of an underlying potential) and a non-conservative current (which might induce stationary persistent currents at long times). The non-conservative part leads to the fact that the dynamics of SGD	may show oscillations, and these oscillations may even prevent the algorithm from converging to the 'right' local optima. The theoretical analysis is carried-out very nicely, and the theory is supported by experiments on two-dimensional toy examples, and Fourier-spectra of the iterates of SGD.

This is a nice paper which I would like to see accepted. In particular I appreciate that the authors stress the importance
of 'non-equilibrium physics' for understanding the SGD process. Also, the presentation is quite clear and the paper well written.

There are a few minor points which I would like to ask the authors to address:

1. Why cite Kingma and Welling as a source for variational inference in	section 3.1? VI is a much older	field, and Kingma and Welling proposed a very special form of VI, namely amortized VI with inference networks. A better citation would be Jordan et	al 1999.

2. I'm not sure how much to trust the Fourier-spectra. In particular, perhaps the deviations from Brownian motion could also be due to the discrete	nature of SGD (i.e. that the continuous-time formalism is only an approximation of a discrete process). Could you elaborate on this?

3. Could you give the reader more details on how the uncertainty estimates on the Fourier transformations were obtained?

Thanks.

---

> ### Author Response · Authors · 2017-12-12
> **Reply to Rev. 3**
>
> Thank you for your comments. Please read below for our clarifications.
>
> >>Why cite Kingma and Welling for variational inference, e.g., cite Jordan ‘99, VI is a much older field
>
> Good point: we will also include older citations.
>
> >>Not sure how much to trust Fourier spectra, deviations from Brownian motion can also be due to discretization
>
> Note that the plot in Fig. 3a is the discrete Fourier transform of (x_{k+1} - x_k)_k. The trajectory is of length 10^5 epochs and sampled at each epoch; we are thus sampling at a very high frequency, well above the Nyquist rate. Low frequency modes in the continuous-time dynamics will not be affected by such a discretization, high frequency modes might, see the right part of Fig. 3a.
>
> The FFT, which is expected to be flat for Brownian motion, is distinctly non-flat in our experiments. This result is also predicted by other experiments in Sec. 4.1 and Fig. 3b, and our theoretical results. So the Fourier spectra are just one more confirmation of the claim.
>
> >>Give more details on how the uncertainty estimates on the Fourier transformations were obtained
>
> This is described in the caption of Fig. 3. The FFT is computed, independently, for the one-dimensional trajectory of each weight. The standard deviation across all the weights is depicted as the “error band”. The eigenmodes of the weight vector are also the eigenmodes of the trajectory of each weight; it is indeed surprising that different weights have very similar amplitude.

---

### Official Review · AnonReviewer2 · 2017-11-27
**Well written, but lacking in novelty.**

**Rating:** 5
**Confidence:** 4

**Review:**

The authors discuss the regularized objective function minimized by standard SGD in the context of neural nets, and provide a variational inference perspective using the Fokker-Planck equation. They note that the objective can be very different from the desired loss function if the SGD noise matrix is low rank, as evidenced in their experiments.

Overall the paper is written quite well, and the authors do a good job of explaining their thesis. However I was unable to identify any real novelty in the theory: the Fokker-Planck equation has been widely used in analysis of stochastic noise in MCMC samplers in recent years, and this paper mostly rephrases those results. Also the fact that SGD theory only works for isotropic noise is well known, and that there is divergence from the true loss function in case of low rank noise is obvious. Thus I found most of section 3 to be a reformulation of known results, including Theorem 5 and its proof.

Same goes for section 5; the symmetric- anti symmetric split is a common technique used in the stochastic MCMC literature over the last few years, and I did not find any new insight into those manipulations of the Fokker-Planck equation from this paper.

Thus I think that although this paper is written well, the theory is mostly recycled and the empirical results in Section 4 are known; thus it is below acceptance threshold due to lack of novelty.

---

> ### Author Response · Authors · 2017-12-12
> **Reply to Rev. 2**
>
> Thank you for your comments. Please see our clarifications below.
>
> >>Unable to identify any novelty in the theory, reformulation of known results, empirical results are known
>
> We are glad to help:
> 1. While it is widely *believed* that SGD acts as an “implicit regularizer”, to the best of our knowledge we are first to *prove* that it performs variational inference: SGD minimizes an average potential along with an entropic regularization term.
> 2. While someone may have noticed that mini-batch noise in deep networks is highly non-isotropic, nobody had connected this to convergence properties of SGD for deep nets.
> 3. The fact that anisotropy in deep networks causes the potential Phi to be different than the function upon which SGD evaluates its gradients was *not known*, nor proven, before.
> 4. The fact that the most likely trajectories of SGD for deep nets are limit cycles was *not known*, nor proven.
> We have scouted the literature diligently, but of course it is possible that we may have missed work where any of the above empirical and theoretical results may have been described. We will gladly examine specific references if provided.
>
> >>Fokker-Planck equation has been widely used before
>
> We surely do not claim to be the first to use the Fokker-Planck equation; it is a standard tool in the analysis of stochastic processes.
>
> >>Fact that SGD theory only works for isotropic noise is well-known, that there is divergence from the true loss is obvious
>
> The issue is not that there is “divergence from the true loss”, but precisely of what *nature* it is. To the best of our knowledge, we are the first to point out -- and prove -- that SGD for deep nets has limit cycles as its most likely trajectories. This is surely not obvious: in fact, most of the literature focuses on which *critical points* SGD converges to. We show that, with anisotropic noise, it converges to none. Quite non-obvious, frankly.
>
> >>Common technique in stochastic MCMC, did not find any new insight into manipulations
>
> MCMC theory constructs grad f and D given a log-likelihood Phi that one would like to draw samples from. This paper is about the reverse direction: given a grad f and a D, what is the Phi? This is a novel question and pertinent to understanding the efficacy of SGD for deep networks; it is not under the purview of the MCMC literature. We *decompose* grad f into symmetric and anti-symmetric terms and develop assumptions and theory that enables us to do so.
>
> To emphasize, MCMC methods start with a given Phi, whereas we find the Phi. The two are completely opposite directions, even if some formulae might look familiar from the MCMC literature.

---

### Official Review · AnonReviewer1 · 2017-11-28
**An interesting paper on analyzing the impact of gradient noise for SGD**

**Rating:** 6
**Confidence:** 4

**Review:**

This paper develop theory to study the impact of stochastic gradient noise for SGD, especially for deep neural network models. It is shown that when the gradient noise is isotropic normal, SGD converges to a distribution tilted by the original objective function. However, when the gradient noise is non isotropic normal, which is shown common in many models especially in deep neural network models, the behavior of SGD is intriguing, which will not converge to the tilted distribution by the original objective function, sometimes more interestingly, will converge to limit cycles around some critical points of the original objective function. The paper also provides some hints on why using SGD can get good generalization ability than gradient descend.

I think the finding of this paper is interesting, and the technical details are correct. I still have the following comments.

First, Assumption 4 seems a bit too abstract. It is not easy to see what the assumption means. It would be better if an example is given, which is verified to satisfy the assumption.

Another comment is related to the overall content of this paper. Thought the paper point out that SGD will have the out-of-equilibrium behavior when the gradient noise is non isotropic normal, it remains to show how far away this stationary distribution is from the original distribution defined by the objective function.

---

> ### Author Response · Authors · 2017-12-12
> **Reply to Rev. 1**
>
> Thank you for your comments. Please see our clarifications below.
>
> >>Assumption 4 seems a bit too abstract, can you give an example
>
> Example 13 illustrates the effects of the assumption; we will point the readers to it in Sec. 3. Another example is in three dimensions, where the assumption is akin to Helmholtz decomposition of a vector field into divergence-free and curl-free components. We allow the force j(x) to be non-trivial, j(x) neq 0 corresponds to broken detailed balance while j(x) = 0 corresponds to detailed balance. This assumption is motivated by the second-law of thermodynamics as discussed in Appendix B.
>
> >>How far away is the stationary distribution from the original one
>
> The relation between the two is the offset described in Thm. 17. This difference scales linearly with learning rate/batch-size; which can be large in practice because deep networks are trained with small batch-sizes and/or large learning rates. The divergence of the matrix Q is also explicitly computable, see (A13) and Remarks 19-20. Doing so is however computationally challenging for large networks, and a subject of our future investigation.

---

### Public Comment · ~James_T_Griffin1 · 2017-12-02
**A mathematician's perspective**

I really like this paper and have learnt a lot from reading it.  I think the basic ideas behind it are very important indeed and I don't know of anywhere else they are written down.  However I think it has some major issues.

Most importantly I think the statements at the very start the introduction "[Φ] is only a function of the architecture and the dataset" and at the start of Section 3, "The potential Φ(x) depends only on the full-gradient and the diffusion matrix, and will be made explicit in Section 5." are *wildly* misleading.  They depend on both of Assumptions 4 and 16, which even at the start of Section 3 have not been made yet.  I think's it's entirely reasonable to think that "in the wild" Φ(x) would depend on the learning rate, and the burden of proof to convince a reader otherwise should be very high.

I am also confused by the use of the term "full-gradient".  In Lemma 14 formula for Φ involves U, but U depends on the Hessian of f.  So more than the gradient of f at x.

The very first equation of the introduction is a tautology if Phi is defined as in equation (6) and only has value if it has a given formula which is never actually given in the text and only alluded to (I don't count Lemma 14 as that applies to a quadratic form only).  There is nothing logically wrong about doing this and I personally find it quite entertaining, but it does feel like a sleight of hand that could hide the assumptions from an inattentive reader.

The second part of Theorem 5 is just an entropy maximisating theorem which is in every standard textbook (eg it's a corollary of Thm 12.1.1 from Elements of Infromation Theory 2nd Ed. by Cover and Thomas).  The first part feels familiar but I couldn't point you to a specific reference.

Concerning Assumption 4... This assumption is not invariant w.r.t. a change in coordinates of the parameter space.  So it is reliant on the Euclidean metric, but why not any other metric, perhaps the Fisher metric?  In physics there is usually some kind of symmetry group on the underlying space pushing us to a metric, but there isn't here so I don't think the argument given backing up this assumption is very convincing.  In fact my opinion is that this assumption is wrong (though I don't think this disqualifies it from being assumed, it's interesting enough to see what happens given the assumption).

Finally in Section 4, the experimental section about Brownian motion, I don't think the null hypothesis that SGD undergoes Brownian motion at a local minimum (which I assume is approximated by a quadratic form) is very strong.  Is the evidence consistent with Brownian motion in a degenerate minimum with more complicated topology?

So in summary I really like this paper, but it needs both the assumptions 4 and 16 to be prominent.  To my mind neither of the assumptions are strictly correct, but that doesn't disqualify them from being made or stop the resulting models being taken seriously.

---

> ### Author Response · Authors · 2017-12-05
> **Re: A mathematician's perspective**
>
> Thank you for your comments. Our responses are enclosed below.
>
> 1. “assumption 4 is not invariant w.r.t. a change in coordinates of the parameter space”, “in physics there is usually some kind of symmetry group”
>
> We do not know of general results that indicate symmetries in SGD dynamics which would suggest the “right” metric space to perform our analysis. Indeed, our results indicate that such an analysis would be promising because this metric is expected to depend upon the architecture.
>
> 1.1 “don't think the argument given backing up assumption 4 is very convincing”, “my opinion is that it is wrong (but don't think this disqualifies it from being assumed)”
>
> Assumption 4 is motivated by an argument that interprets the Fokker-Planck equation as a physical system in contact with the environment through energy exchange of the diffusion term. This assumption is sufficient to ensure that the second law of thermodynamics holds for such a system and is standard in the analysis of irreversible processes, see [1, 2, 3]. The second law may be violated when considering a few molecules of a gas, or analogously, a few trajectories of SGD, but our results always deal with the entire steady-state distribution.
>
> 2. "[Φ] is only a function of the architecture and the dataset is wildly misleading", “They depend on both of Assumptions 4 and 16”, entirely reasonable to think that in the wild Φ(x) would depend on the learning rate”
>
> One only needs assumption 4 to ensure that Phi does not depend on beta. The proof follows from (A4). Define Phi(x) = -beta^{-1} log rho^ss_beta(x) and J^ss_beta from Appendix D accordingly, we have used the subscript to emphasize the dependence on beta. (A4) implies that J^ss_1 is orthogonal to grad rho^ss_1. Decompose -grad f(x) again as
>              -grad f = J_1^ss/rho_1^ss - D grad (log rho^ss_1)
>                          = (rho_1^ss)^{-beta} ((rho_1^ss)^{beta-1} J_1^ss) - beta^{-1} D grad (log rho_1^ss)^beta.
> Now note that div((rho_1^ss)^{beta-1} J_1^ss) = 0 by assumption 4, this lets us identify J_beta^ss = J_1^ss (rho_1^ss)^{beta-1} and rho_beta^ss = (rho_1^ss)^{-beta}. The later gives the result that Phi does not depend on beta.
>
> To conclude, under assumption 4, Phi does not depend on the learning rate or the batch-size, it is only a function of the architecture and the dataset. Also see #3 below, for isotropic noise, Phi(x) = f(x) without any assumptions.
>
> 3. “The very first equation of the introduction is a tautology if Phi is defined as in equation (6)” / “feels like a sleight of hand that could hide the assumptions” / “The first part feels familiar”
>
> Indeed, the minimizer of (11) is of the form (6). However, the key point of Thm. 5 is instead that the Fokker-Planck equation reaches this minimizer *monotonically*. This is far from a sleight of hand, and if gradient noise is isotropic, in complete rigor, (11) with Phi = f becomes the celebrated Jordan-Kinderleher-Otto (JKO) functional [4]; we have steepest descent in the Wasserstein metric in this case, in addition to monotonic decrease. The JKO functional is one of the major results of the theory of optimal transportation in the 20th century, see Sec. 4.3 in [5]. The implicit definition of Phi in (6) is only used for Thm. 5. We give a completely explicit formula for Phi, in terms of f(x) and D(x), in Thm. 17 and (A13).
>
> 4. “needs both the assumptions 4 and 16 to be prominent. To my mind neither of the assumptions are strictly correct, but that doesn't disqualify them from being made or stop the resulting models being taken seriously.“
>
> We will make these assumptions prominent in the introduction. In our opinion, assumption 4 is mild and the low frequency modes of the FFT in Fig. 3a already validate it. Assumption 16 is less mild, but it is widely used by physicists and biologists (we provide references in the paper) to study real systems where it has been seen to hold.
>
> 5. “does SGD undergo Brownian motion near a minimum”, “Is the evidence consistent with Brownian motion in a degenerate minimum with more complicated topology?”
>
> Irrespective of the topology, for isotropic noise, at low enough temperature, SGD performs Brownian motion near a minimum up to the first order. This can be seen from (3).
>
> [1] Prigogine, I. (1955). Thermodynamics of irreversible processes, volume 404. Thomas.
> [2] Qian, H. (2014). The zeroth law of thermodynamics and volume-preserving conservative system in equilibrium with stochastic damping. Physics Letters A, 378(7):609–616.
> [3] Frank, T. D. (2005). Nonlinear Fokker-Planck equations: fundamentals and applications. Springer Science & Business Media.
> [4] Jordan, R., Kinderlehrer, D., and Otto, F. (1997). Free energy and the Fokker-Planck equation. Physica D: Nonlinear Phenomena, 107(2-4):265–271.
> [5] Santambrogio, F. (2017). Euclidean, metric, and Wasserstein gradient flows: an overview. Bulletin of Mathematical Sciences, 7(1):87–154.

---

### Author Response · Authors · 2018-01-05
**Updates to the paper**

We thank the reviewers and the commentators for their feedback, individual clarifications are enclosed below. We have updated the paper to incorporate these inputs, the summary of changes is as follows.

1. The diffusion matrix D(x) depends on the current iterate of SGD. This implies that the stochastic process be interpreted in the Ito sense. The Fokker-Planck equation (FP) in Lemma 2 was written in the Fick’s Law form earlier with a term div (D grad rho). This is now changed to the Ito form with the term div (div (D rho)).

2. The above change results in an extra term beta^{-1} div(D) in the definition of the conservative force in (8). All results in our paper remain unchanged upon consistently adding this term.

3. Corollary 8 shows that if the diffusion matrix is identity, Theorem 5 recovers the Jordan-Kinderleher-Otto (JKO) functional in optimal transportation. Trajectories of the Fokker-Planck equation perform steepest descent in the Wasserstein metric on (11) in this case.

4. We have rewritten Theorem 20 to make it more precise.

5. Upon the suggestion of Reviewer 3, we have added older references for variational inference; see Remark 10.

6. We point the readers to Example 17 in Assumption 4 for an illustration.

---

### Decision · Program_Chairs · 2018-01-29
**ICLR 2018 Conference Acceptance Decision**

**Decision:**

Accept (Poster)

**Comment:**

Dear authors,

Based on the comments and your rebuttal, I am glad to accept your paper at ICLR.